# Knowledge of Human Monkeypox Infection among Final Year Medical, Pharmacy, and Nursing Students: A Multicenter, Cross-Sectional Analysis from Pakistan

**DOI:** 10.3390/healthcare11202777

**Published:** 2023-10-20

**Authors:** Sahabia Masood, Noorah A. Alkubaisi, Maryam Aslam, Muhammad Salman, Mohamed A. Baraka, Zia Ul Mustafa, Tauqeer Hussain Mallhi, Yusra Habib Khan, Johanna C. Meyer, Brian Godman

**Affiliations:** 1Department of Medicine, University Medical and Dental College Faisalabad, Faisalabad 38000, Pakistan; sahabiamasood15@gmail.com; 2Department of Botany and Microbiology, College of Science, King Saud University, Riyadh 11451, Saudi Arabia; nalkubaisi@ksu.edu.sa; 3Basic Health Unit (BHU), 554GB, Tehsil Tandlianwala, Faisalabad 38000, Pakistan; maryamaslam00999@gmail.com; 4Institute of Pharmacy, Faculty of Pharmaceutical and Allied Health Sciences, Lahore College for Women University, Lahore 54000, Pakistan; msk5012@gmail.com; 5Department of Pharmacy, Fatima College of Health Sciences, Abu Dhabi P.O. Box 3798, United Arab Emirates; 6Clinical Pharmacy Department, Faculty of Pharmacy, Al-Azhar University, Cairo 11651, Egypt; 7Discipline of Clinical Pharmacy, School of Pharmaceutical Sciences, Universiti Sains Malaysia, Gelugor 11800, Penang, Malaysia; 8Department of Pharmacy Services, District Headquarter (DHQ) Hospital, Pakpattan 57400, Pakistan; 9Department of Clinical Pharmacy, College of Pharmacy, Jouf University, Sakaka 72388, Saudi Arabia; thhussain@ju.edu.sa (T.H.M.); yhkhan@ju.edu.sa (Y.H.K.); 10Department of Public Health Pharmacy and Management, School of Pharmacy, Sefako Makgatho Health Sciences University, Pretoria 0208, South Africa; hannelie.meyer@smu.ac.za (J.C.M.); brian.godman@strath.ac.uk (B.G.); 11Strathclyde Institute of Pharmacy and Biomedical Science (SIPBS), University of Strathclyde, Glasgow G4 0RE, UK; 12Centre of Medical and Bio-Allied Health Sciences Research, Ajman University, Ajman P.O. Box 346, United Arab Emirates

**Keywords:** outbreak, monkeypox, medical, knowledge, healthcare students, Pakistan

## Abstract

The World Health Organization (WHO) declared the monkeypox outbreak a public health emergency in June 2022. In Pakistan, positive cases of monkeypox were reported in April 2023. Healthcare workers (HCWs) are considered as a front-line force to combat such outbreaks. A questionnaire-based cross-sectional study was conducted among 11 public sector educational institutions in Punjab, Pakistan, during May and June 2023 among final year medical, pharmacy, and nursing students concerning their knowledge of monkeypox. This included the signs/symptoms of monkeypox. Healthcare students were chosen as they are the HCWs of tomorrow. A total of 389 healthcare students participated in the study, with a mean age of 23.17 ± 1.72 years, and the majority were female. The mean knowledge score was 17.69 ± 4.55 (95% CI 17.24–18.14) out of a maximum total knowledge score of 26 (each correct answer was given a score of 1). The proportion of students with good, moderate, and poor knowledge was 21.6%, 43.2%, and 35.2%, respectively. Age (*p* = 0.017), gender (*p* < 0.001), and education (*p* < 0.001) had a significant impact on the knowledge score. In the multivariate linear regression model, education was the only significant factor linked to knowledge scores. Overall, the majority of future HCWs had moderate knowledge of monkeypox. Consequently, educational activities are needed to improve monkeypox-related knowledge among future HCWs. Furthermore, emerging infectious diseases should be routinely incorporated into HCW curricula.

## 1. Introduction

The monkeypox (Mpox) virus was discovered in Copenhagen, Denmark, in 1958 in cynomolgus monkeys during two outbreaks of smallpox [1,2]. From 1958 to 1968, many outbreaks of monkeypox were reported in colonies of monkeys in the United States and the Netherlands, including deaths, although no cases of infection in humans were reported [3]. The first case of human Mpox was reported in 1970 in a 9-month-old child during smallpox surveillance in the Democratic Republic of the Congo (DRC) [4]. The baby patient presented with otitis, mastoiditis, painful cervical lymph nodes, and skin lesions, from which the Mpox virus was isolated. Although the baby recovered from Mpox, he later succumbed to complications due to measles before discharge from hospital [4]. Between 1970 and 1971, six additional positive cases of Mpox were reported in humans in West African countries, particularly among young children who were not vaccinated against smallpox [5]. Mpox was regarded as an African disease with sporadic cases reported throughout Africa until 2003, when the first case of Mpox was reported in the United States among rats. From rats, the virus was transmitted to dogs and subsequently to humans, mainly young adults and children [6]. From 2017 to 2018, a large monkeypox outbreak was reported in Nigeria and five infected cases of Mpox were reported in the United Kingdom, Israel, and Singapore in patients with travel history in Nigeria [7,8,9,10]. In May 2022, another outbreak of Mpox occurred and a number of positive cases were documented in the United Kingdom, Portugal, and Italy [11,12,13]. The World Health Organization (WHO) subsequently declared the recent Mpox outbreak a public health emergency on 23 July 2022 [14]. More than 87,000 positive cases, along with 130 deaths, have been reported due to the Mpox outbreak in 111 countries as of 27 April 2023 [15]. To date, most of the deaths due to Mpox were reported in the American region followed by the African region [15].

Mpox is a viral zoonotic illness caused by the monkeypox virus, a species of the Orthopoxvirus genus [16,17,18]. Two distinct clades of Mpox have been identified, one is Clade I, previously known as the Congo Basin or central African clade, and the other is Clade II, the former West African clade [16,19]. Overall, the Congo Basin clade (Clade I) has a higher case fatality rate [6,20,21]. Being zoonotic, evidence of Mpox virus infection has been reported in animals and, as mentioned, it has been transmitted from animals to humans [1]. Mpox can be transmitted from one person to another through direct or indirect contact with a rash and different bodily fluids [16,22,23,24,25]. Direct contact with clothing, bedding, and towels, as well as eating from the same dishes as an infected person, can transmit the disease to others [16,20,23,25,26]. Moreover, the virus can also spread from pregnant women to the fetus through the placenta, or from an infected parent to a child during or after birth through direct contact [27].

In the multi-country Mpox outbreak of 2022, the majority of cases (98%) were observed in bisexual men, with greater prevalence among immunocompromised individuals, mainly those with HIV [28,29]. In more than two-thirds of cases, transmission took place through sexual contact, with negligible transmission through physical contact (4.6%) [30]. Overall, pregnant women, children, and immunocompromised patients are at higher risk of developing Mpox [31].

The most common symptoms among positive cases of Mpox are a generalized or genital rash, fever, chills, swollen lymph nodes, headaches, muscle and joint aches, exhaustion, and respiratory symptoms like sore throats, coughs, and nasal congestion [15,32,33]. Skin rashes and lesions are the key characteristics of the skin eruption phase of Mpox, and are typically more localized on the face in up to 95% of cases [33,34,35,36,37]. In most cases, symptoms disappear on their own; however, severe cases may develop complications, including skin infections, pneumonia, confusion, eye diseases, proctitis, and painful urination, and may lead to death in up to 10% of cases [31].

The management of Mpox includes mainly supportive symptomatic care; however, antiviral drugs including tecovirimat, cidofovir, and brincidofovir are used for symptomatic treatment of severe or immunocompromised patients [17,38]. Two vaccines are available for use in the current outbreak (ACAM2000 and MVA-BN) [39]. However, there needs to be a careful balance between the risks and benefits of vaccination [37,40,41]. Smallpox vaccines have also been observed to be at least 85% effective in preventing MPox [40].

In Pakistan, the very first case of Mpox was confirmed by the National Institute of Health (NIH), Islamabad, on 25 April 2023 in a 25-year-old man with travel history in Saudi Arabia [42]. Moreover, as per local media reports, four more suspected cases of Mpox were isolated at government quarantine centers near Karachi airport; among these, two had travel history in Somalia [43]. Another recent case of Mpox was reported in Karachi in a 36-year-old man, again coming from Saudi Arabia, on the 5 May 2023 [44]. The increasing number of human Mpox cases at this time necessitated the importance of prevention, early detection, and rapid reporting, as well as quick and effective management from healthcare workers (HCWs) in designated health facilities. In this respect, the Health Services in Sindh and Punjab issued high alert advice to all HCWs working in public sector hospitals for the quick detection and management of Mpox cases throughout the province. In addition, an isolation ward containing 5–10 beds for Mpox cases was established. Moreover, the NIH, Government of Pakistan, also issued guidelines to prevent the spread of Mpox throughout the country on 26 April 2023 [45].

Taking into account the current global Mpox outbreak, the role of HCWs, as well as healthcare students in terms of their future role, is pivotal for the effectiveness of any treatment and prevention strategy. We have seen in the case of the MMR vaccine and the recent COVID-19 vaccines that concerns with vaccines among HCWs can have a devastating impact on their uptake generally and on future infectious diseases [46,47,48,49,50,51]. Consequently, HCWs play an important role in developing effective and targeted strategies for the prevention of infectious diseases through educating and influencing the population [52,53]. Moreover, they are key players in raising awareness on the ways of infection transmission in addition to treating Mpox-positive patients. However, there have been concerns about their knowledge of Mpox, including among physicians and community pharmacists, across countries [54,55,56,57,58,59]. There are also concerns about the level of knowledge of healthcare students regarding Mpox, as well as associated conspiracy theory issues. Both situations are a concern given the important role of HCWs in optimizing future vaccination campaigns and addressing misinformation, such as conspiracy theories linked with COVID-19 vaccines [60]. Educating patients about Mpox should also enhance their willingness to be vaccinated [61].

To the best of our knowledge, we are unaware of any study specifically in Pakistan on evaluating the knowledge of future HCWs (medical doctors, pharmacists, and nurses) regarding MPox, including its sources, prevention, and management. However, we are aware that medical students from Pakistan were included in a recent study by Abd El Hafeez et al. (2023) [56]. Consequently, we sought to address this evidence gap by conducting a cross-sectional, multicenter study among final year medical, pharmacy, and nursing students to assess their knowledge of the sources of Mpox infection, the signs and symptoms of the infection as well as ways of prevention and management strategies. The findings can be used to guide developments in the curricula of these student HCWs where pertinent.

## 2. Materials and Methods

### 2.1. Study Design, Setting, and Population

A cross-sectional, questionnaire-based study design was employed to collect data from final year medical, pharmacy, and nursing students at different public sector educational institutions in the Punjab province of Pakistan. Punjab was selected for the purpose of this study due to fact that that it contains the majority (~65%) of the country’s population along with greater numbers of the targeted educational institutions [62,63]. We also deliberately targeted public sector universities for this study as they include the widest range of students. According to the Pakistan Medical and Dental Association (PMDC), there are currently 19 operational public sector medical colleges/universities in Punjab, offering Bachelor’s degrees in medicine, MBBS (five years plus one year residency). On average, each of these public sector institutions have between 50 and 100 students in their final year [64,65]. Based on the information provided by a representative from the Pakistan Pharmacy Council (PPC), there are currently seven public sector universities offering a Doctor of Pharmacy (five years) program in Punjab Province with 50–150 final year students [65]. More than 50 public sector nursing institutes offer a graduate program in nursing (four years), and each institution has between 50–80 final year nursing students [65,66]. The current survey was conducted in 11 public sector institutions (3 medical, 2 pharmacy, and 6 nursing institutions).

### 2.2. Survey Instrument

The outcome measures used in our survey were taken from previous monkeypox-related studies conducted in different countries after appropriate permission from the corresponding authors [55,56,67,68,69,70]. The survey instrument was in English. This is because English is the primary medium of instruction in higher education in Pakistan. Consequently, an instrument available in the Urdu language was not considered necessary for the purpose of this study.

Content validation of the survey instrument was performed by a panel of seven experts from the field of medicine, pharmacy, and nursing. Minor revisions were made to the questionnaire based on the comments and suggestions made by the panel. Subsequently, a pilot survey was conducted among 30 students to assess the internal consistency of the study instrument. Cronbach’s alpha was calculated, and the results showed adequate internal consistency of the study instrument (α > 0.7). The final version of the study instrument contained the following sections:**Section I**—contained nine questions related to the demographic characteristics of the study participants, e.g., name of the institution, age, sex, residence, and family income.**Section II**—consisted of 27 items, each having three response options (‘yes’, ‘no’, and ‘don’t know’). This section contained questions about their knowledge of Mpox, signs and symptoms, ways of transmission, and prevention and management of positive cases. Each correct answer was scored “1”, whereas wrong/don’t know responses were scored zero. The scores of all the items were summed to calculate the cumulative knowledge score. This score was used to determine the correct answer rate (number of correct answers by the participant/27 × 100). Bloom’s cut-off criteria were used to categorize students as having “good” (80% to 100% score), “moderate” (60–79% score), and “poor” Mpox-related knowledge (<60% score) [67].**Section III**—enquired about the sources of information used by the study participants, e.g., printed or electronic media, books, journals, peer, and mobile or online applications.

### 2.3. Sample Size Calculation

Sample size calculation for the number of participants to be recruited for the survey was computed using the Raosoft online sample size calculator (http://www.raosoft.com/samplesize.html, accessed on 27 March 2023). Assuming a population size of 15,000 final year medical, pharmacy, and nursing students, and an expected response distribution of 50% to provide the largest sample, at a 95% confidence level and 5% margin of error, the minimum sample size was calculated as 375. However, we increased the sample size by 3–5% to account for any incomplete data.

### 2.4. Inclusion and Exclusion Criteria

Final year medical, pharmacy, and nursing students enrolled at the selected educational institutions who were willing to participate in the survey were included. Students other than those from the selected institutions, those not willing to participate in the study, and those not currently enrolled in the final year of study were excluded from the current study.

### 2.5. Data Collection Procedure

Data were collected over a period of two months (May–June 2023) after the various educational institutions reviewed the study and granted permission to collect data amongst the target population. A team of investigators was first trained in data collection procedures. Following this, the various institutions were contacted to arrange a dedicated date for data collection. Participants were recruited using convenient sampling. Final year students were approached by the investigators in their classes, information about the study was provided, and the objectives of the study were explained. Participants were ensured that no personal information would be collected from them and that they could leave the survey at any stage, should they wish to do so. Those who were willing to participate in the survey were provided with the study questionnaire (Appendix A), with the request to provide their response. After 15–20 min, the study questionnaires were collected from the participants.

### 2.6. Statistical Analysis

Participant responses were coded and entered in Microsoft Excel and then SPSS version 22 for statistical analysis. Continuous data were summarized as means with standard deviations, while categorical data were summarized as numbers (n) and percentages (%).

The Mpox knowledge scores were compared among demographic variables using an independent sample *t*-test and a one-way ANOVA, as appropriate. Furthermore, multiple comparisons were made among the significant trichotomous variables using Tukey’s HSD and/or Games–Howell post hoc tests. Predictors of a better monkeypox knowledge score were assessed in a multiple linear regression analysis. A *p*-value of less than 0.05 was considered statistically significant for all the statistical procedures.

### 2.7. Ethical Considerations

Ethical approval of the current study was obtained from the Office of Research, Innovation and Commercialization (ORIC), Lahore College for Women University (LCWU), Jail Road, Lahore. Moreover, approval of the study was also obtained from the participating institutions. Written informed consent was obtained from all the participants prior to their participation in the study.

## 3. Results

A total of 500 questionnaires were distributed among the potential participants and 389 students provided responses, which gave a response rate of 77.8%. Of the 389 participant students, 165 were medical (42.4%), 78 were pharmacy (20.0%), and 146 (37.6%) were nursing students. The characteristics of the sample population are presented in Table 1. The mean age of sample was 23.17 ± 1.72 years, with the majority of students being between the ages of 21 and 25. There was a predominance of female students (69.9%) and those belonging to urban areas (67.1%).

A quarter (25.4%) of the study participants reported that they had a parent in the medical profession. All the students had heard of Mpox disease. The main source of this information was social media (48.1%), followed by electronic/print media (20.3%).

Participant responses to the questions assessing their knowledge of Mpox are shown in Figure 1. Almost 82% of student HCWs knew Mpox was not a bacterial infection and 79.4% were aware that it was a viral disease. A total of 27.2% of students knew that Mpox and smallpox have similar clinical features, and 57.8% knew monkeypox’s incubation period. Regarding the mode of transmission, 69.4% were aware Mpox can be transmitted from animals to humans via contact with blood, body fluids, and/or lesions. A total of 69.9% of student HCWs knew Mpox can also be transmitted by consuming insufficiently cooked meat. A total of 69.4% knew Mpox can transmitted from human to human through contact with respiratory secretions, skin lesions, or objects contaminated by an infected person.

Participants generally had insufficient knowledge about the signs/symptoms of the disease (correct answer rate of 49.1% to 68.4%). In addition, the study participants had only moderate knowledge about the prevention of Mpox. The majority of students were aware though that Mpox was a self-limiting disease and 75.3% knew that the mainstay of Mpox treatment is symptomatic supportive care. However, 19.8% believed that antibiotics were very effective in treating Mpox.

The mean knowledge score was 17.69 ± 4.55 (95% CI 17.24–18.14) out of a total knowledge score of 26. Overall, the frequency of students with good, moderate, and poor knowledge was 21.6%, 43.2%, and 35.2%, respectively (Figure 2).

Comparisons of knowledge scores between demographic variables revealed a higher age (*p* = 0.017), gender (male 19.97 vs. female 16.71, *p* < 0.001), family income (*p* < 0.001), and education (*p* < 0.001) as having a significant impact on the knowledge score. In a multiple comparison using Tukey’s HSD test (Table 2), students aged < 20 years (‘a’) had significantly lower knowledge than those aged between 21 and 25 years (‘b’ − *p* = 0.024) and >25 years old (‘c’ − *p* = 0.023). However, there was no significant difference in knowledge scores between 21–25-year-old students and >25-year-old students (*p* = 0.641). Furthermore, nursing students were found to have a lower knowledge score than medical (*p* < 0.001) and pharmacy students (*p* < 0.001).

A multiple regression analysis was performed to confirm the predictors of better Mpox knowledge (Table 3). Education was found to be the only significant factor linked to knowledge scores (standardized beta = −0.589, *p* < 0.001). Unstandardized beta values revealed that with every one level/unit increase in the education variable (1 = medical, 2 = pharmacy, and 3 = nursing), the dependent variable (MPoX knowledge score) decreased by nearly 3 units.

## 4. Discussion

We believe this is the first study undertaken in Pakistan to evaluate the level of knowledge of future HCWs in Pakistan about the origin, signs and symptoms, transmission from one person to another, as well as the prevention and management of monkeypox. Moreover, knowledge scores and demographics variables between the different HCW students have been compared. Overall, the majority of the students have good (21.6%) or moderate (43.2%) knowledge scores, with educational level being a significant factor linked with knowledge scores.

We highlighted the resilience and recognition of front-line HCWs in combating the pandemic caused by COVID-19. Consequently, we believe that the awareness and capacity building of future HCWs to handle any pandemic situation is of the utmost importance. After overcoming the COVID-19 pandemic in the past three years, a new zoonotic monkeypox virus outbreak has recently been reported in many non-endemic nations. Although it is an unusual, self-limiting infection that is often mild in most cases, early detection, reporting, and management are crucial for disease control. In order to combat an ongoing outbreak, the general public, high risk groups, and health authorities need to develop appropriate awareness campaigns as well as prevention strategies. Besides this, the role of front-line HCWs and their understanding about the disease and its management are crucial factors, particularly for those patients who develop moderate to severe disease and need hospitalization. Moreover, HCWs are primarily involved in the identification of confirmed or suspected cases as well as educating the public about preventive measures. Consequently, their full understanding of the disease, including its prevention and treatment, is of vital significance to address infectious diseases like the recent Mpox outbreak in the country.

Our study revealed that the majority of future HCWs are aware of the origin of Mpox in terms of its viral nature; however, a significant proportion of student HCWs were unaware of the primary occurrence of Mpox, i.e., typically in rainforest areas of Africa. These findings are similar to a previous study in Saudi Arabia, in which the majority of physicians did not have knowledge about the occurrence of Mpox in the African region [68], with similar concerns about the knowledge of Mpox among physicians in other countries [57,58,59]. Another study conducted in Malaysia reported similar findings about knowledge of the origin of Mpox among preclinical and clinical dental students [71], mirroring knowledge concerns among student HCWs in other countries [54,55,56]. Future HCWs must be taught about infectious diseases including Mpox in terms of their history, origin, and primary occurrence. This will prepare them to better handle future outbreaks and address misinformation, especially any misinformation proliferated via social media platforms.

Encouragingly, more than two -thirds of the participating student HCW population in our study provided correct answers about the transmission of the Mpox virus from animals to humans, and between humans through bodily fluids, respiratory secretions, and contaminated objects. However, a study in the United Arab Emirates (UAE) indicated that a higher proportion of university students, including medical students, knew about the ways of transmission of MPox from animal to human and human to human [67]. Encouragingly though, our findings are better than the findings of a previous study conducted in Jordan, where only 37% of health school students knew about the transmission of Mpox from one person to another [54]. These findings indicate that awareness sessions and seminars should be initiated among future HCWs to raise their knowledge about the transmission of the Mpox virus, particularly to ensure they are ready to educate patients where pertinent.

Our survey revealed that more than half to two-thirds of the study population were well aware about the signs and symptoms of Mpox, including skin rashes, pustules, and lymphadenopathy. These findings are similar to a study reported in Saudi Arabia, where a similar percentage of medical students provided the correct answers about the signs and symptoms of Mpox [56]. Overall, healthcare institutions must include a particular subject covering prevalent infectious diseases in the region for better management of patients once qualified.

Appreciably more than 80% of our study participants believed that sufficient hand hygiene with soap or alcohol-based detergents is an effective way to prevent the spread of Mpox. Other preventive measures reported in our survey were avoiding contact with wild animals as well as people with skin rashes, in addition to prompt reporting of any suspected case to local authorities. These findings are similar to the findings of a previous study conducted in 27 countries, where study participants agreed that avoiding contact with wild animals and people with skin rashes, along with rapidly reporting any suspected case to the local health authority, could help prevent Mpox [56].

Our study also revealed that the majority of future HCWs considered supportive therapy, including antipyretics, as helpful with symptomatic relief, with a very small proportion believing the use of antibiotics will relieve symptoms. However, the use of antiviral drugs in the management of severe or hospitalized cases of the Mpox virus is considered an effective management strategy. In line with the findings of our study, previous studies have also reported the use of antiviral drugs during the management of monkeypox cases [45,46].

We are aware of a number of limitations with our study. Firstly, we collected data from only 11 public sector educational institutions in Punjab, with a limited number of participants, with no inclusion of private sector institutions, for the reasons stated. Consequently, we are unable to fully generalize our findings. We also only collected data through convenient sampling; consequently, bias may be associated with our results. Moreover, we were unable to obtain accurate data regarding the Mpox cases reported in Pakistan, including their transmission, signs, symptoms, etc. However, despite these limitations, we believe the findings of our study will be useful for clinicians, health authorities, and policy makers to provide details of baseline information currently possessed by future HCWs regarding infectious disease outbreaks, including Mpox virus, and the implications going forward. This includes potential refining of the curricula of student HCWs.

## 5. Conclusions and Next Steps

Most of the future HCWs in our study possessed moderate to good knowledge regarding the Mpox virus, with just under 35% having poor knowledge. Medical and pharmacy students had better knowledge compared to nursing students. In view of our findings, universities in Pakistan need to review their current curricula regarding infectious diseases and refine these where pertinent. The objective is to fully prepare the HCWs of tomorrow to effectively deal with future outbreaks, including combating misinformation regarding viruses and vaccines, and their implications. This is important given concerns regarding the knowledge of HCWs regarding Mpox and the consequences. We will continue to monitor the situation in Pakistan.

## Figures and Tables

**Figure 1 healthcare-11-02777-f001:**
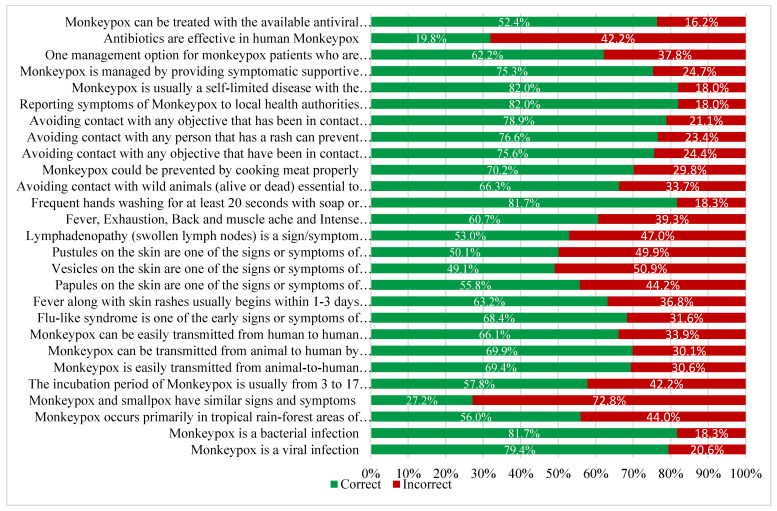
Monkeypox-related knowledge of the study participants (n = 389).

**Figure 2 healthcare-11-02777-f002:**
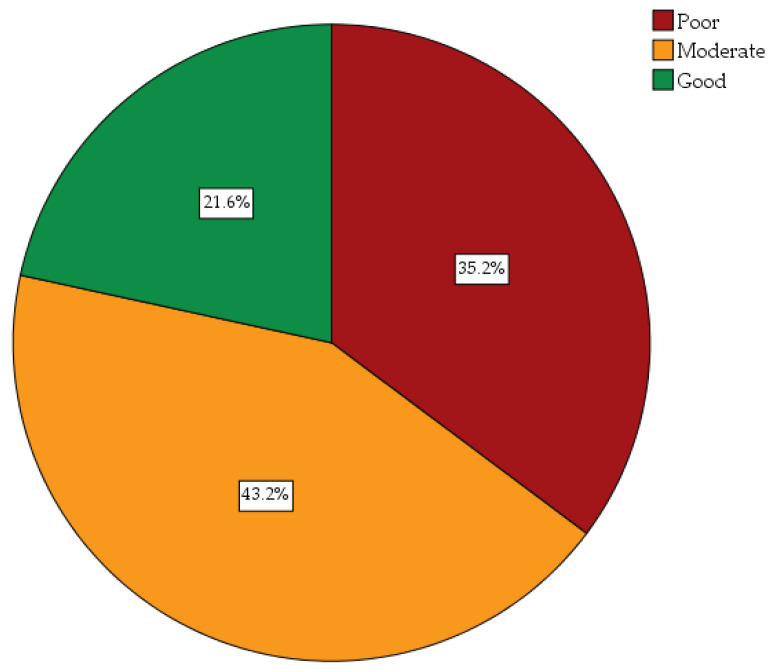
Overall monkeypox-related knowledge scores of study participants.

**Table 1 healthcare-11-02777-t001:** Sample characteristics (n = 389).

Variable	N (%)
**Age**	
≤20 years	30 (7.7)
21–25 years	327 (83.8)
>25 years	33 (8.5)
**Sex**	
Male	117 (30.1)
Female	272 (69.9)
**Education**	
Medical	165 (42.4)
Pharmacy	78 (20.1)
Nursing	146 (37.5)
**Family income (PKR)**	
<25,000	6 (1.5)
25,000–75,000	111 (28.5)
>75,000	272 (69.9)
**Residence**	
Rural	128 (32.9)
Urban	261 (67.1)
**Parent’s profession**	
Medical	99 (25.4)
Non-medical	290 (74.6)
**Received information related to monkeypox in the recent 6 months**	
Yes	389 (100)
No	--
**Source of information**	
Print and electronic media	79 (20.3)
Social media	187 (48.1)
Smartphone application	48 (12.3)
Journals/books	1 (0.3)
Peers	42 (10.8)
Professor/teacher/healthcare providers	32 (8.2)

**Table 2 healthcare-11-02777-t002:** Differences in monkeypox knowledge scores among demographic variables.

Variables	Knowledge Score (Mean ± SD)	*p*-Value	Post Hoc Analysis
**Age (years)**			
≤20 21–25 >25	15.53 ± 4.91 17.80 ± 4.52 18.55 ± 3.99	**0.017**	b > a (*p* = 0.024) c > a (*p* = 0.023) *
**Gender**			
Male Female	19.97 ± 3.04 16.71 ± 4.74	**<0.001**	--
**Family income (PKR)**			
<75,000 >75,000	15.76 ± 4.90 18.52 ± 4.13	**<0.001**	--
**Education**			
Medical Pharmacy Nursing	20.06 ± 3.07 20.04 ± 2.87 13.76 ± 3.93	**<0.001**	a > c (*p* < 0.001) b > c (*p* < 0.001) **
**Residence**			
Rural Urban	17.27 ± 5.16 17.90 ± 4.21	0.228	--
**Parent’s profession**			
Medical Non-medical	18.03 ± 4.78 17.58 ± 4.47	0.391	--

* Tukey HSD test; ** Games–Howell post hoc test; Significant *p*-values in Bold; PKR—Pakistani Rupee.

**Table 3 healthcare-11-02777-t003:** Factors associated with higher monkeypox knowledge scores.

Covariates	Unstandardized Coefficient		Standardized Coefficient	95% CI for B		*p*-Value
	B	SE	Beta	Lower Bound	Upper Bound	
Age	0.445	0.464	0.039	−0.467	1.357	0.338
Gender	0.644	0.451	0.065	−0.243	1.531	0.154
Family income	−0.157	0.453	−0.016	−1.047	0.733	0.729
Education	−3.030	0.255	−0.595	−3.532	−2.528	**<0.001**
Residence	−0.275	0.397	−0.028	−1.055	0.505	0.489
Parent’s profession	0.550	0.426	0.053	−0.288	1.387	0.198

Dependent variable = knowledge score; CI—confidence interval; SE—standard error; Significant *p*-values in Bold.

## Data Availability

Additional data are available from the corresponding authors upon reasonable request.

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
