# Peer review of "Knowledge of Human Monkeypox Infection among Final Year Medical, Pharmacy, and Nursing Students: A Multicenter, Cross-Sectional Analysis from Pakistan"

_healthcare, 2023, doi:10.3390/healthcare11202777_

Round 1
Reviewer 1 Report
The manuscript (ID: healthcare-2621169) aimed to assess the knowledge about Mpox infection (i.e. about the sources, the signs and symptoms, ways of prevention and management of this infection) among final year medical, pharmacy and nursing students at different public sector educational institutions in the Punjab province of Pakistan.
Most sections of this paper are clearly written, but missing information about Mpox infection needs to be added to make the paper more informative for the topic it covers. Above all, there is a lack of detailed information on the multi-country Mpox outbreak in 2022 as a public health emergency on July 23, 2022, as well as on cases of Mpox infection in Pakistan.
Comments (Major revision):
- Line 70: It is mandatory to add a new paragraph that will describe in detail:
ü Socio-demographic characteristics of patients and those who died in the multi-country outbreak of Mpox in 2022.
ü List the populations that were most at risk of Mpox in this outbreak.
ü List the risk factors for Mpox.
ü List the classification according to the level of risk for Mpox infection for exposed persons.
ü Cite appropriate references.
- Lines 105-111: The mentioned cases of Mpox infection in Pakistan were probably surveyed and provided their socio-epidemiological anamnesis. List all known risk factors for Mpox infection that have been identified in those cases. Pay attention to the way the infection is transmitted, personal medical history of the cases (immunosuppression, cancer, HIV, sexually transmitted diseases, etc.).
- The Methods section is satisfactorily written, including most of the subsections presented. As a whole, the appropriate methodology was applied in this paper. But:
ü Lines 161-166: Explain why other questions were not included in the survey. Why did the authors not include in the survey other important questions about Mpox infection, based on their own clinical/epidemiological/public health knowledge and/or work with Mpox cases.
- The Results section presented the data of this research in a satisfactory manner.
- Lines 67-75. Other limitations of this study were not satisfactorily included. As limitations of this study, it is necessary to discuss the sample size, the incompleteness or absence of questions included in this survey about the characteristics of Mpox infection, there are no questions about risk factors and vaccines against Mpox infection.
In summary: This work is not informative enough for the topic it deals with. Neither the word `risk factors' nor `risk exposure' for Mpox infection is mentioned in the paper. Information about this should be entered in the Introduction section. Without such information, about which no question was asked to the respondents in this study, as well as no question about the use of vaccines in the prevention of Mpox infection, this paper is not satisfactorily informative.
The quality of English language is appropriate.
Author Response
Quality of English Language
( ) I am not qualified to assess the quality of English in this paper
( ) English very difficult to understand/incomprehensible
( ) Extensive editing of English language required
( ) Moderate editing of English language required
(x) Minor editing of English language required
( ) English language fine. No issues detected
Author comments: Thank you for this. The paper has now been updated by one of the co-authors who is a native English speaker with over 500 publications in peer-reviewed Journals since 2008. We trust this is now OK.
|
Yes |
Can be improved |
Must be improved |
Not applicable |
|
|
Does the introduction provide sufficient background and include all relevant references? |
( ) |
( ) |
(x) |
( ) |
|
Are all the cited references relevant to the research? |
( ) |
( ) |
(x) |
( ) |
|
Is the research design appropriate? |
( ) |
(x) |
( ) |
( ) |
|
Are the methods adequately described? |
( ) |
(x) |
( ) |
( ) |
|
Are the results clearly presented? |
( ) |
(x) |
( ) |
( ) |
|
Are the conclusions supported by the results? |
( ) |
(x) |
( ) |
( ) |
Comments and Suggestions for Authors
1)The manuscript (ID: healthcare-2621169) aimed to assess the knowledge about Mpox infection (i.e. about the sources, the signs and symptoms, ways of prevention and management of this infection) among final year medical, pharmacy and nursing students at different public sector educational institutions in the Punjab province of Pakistan.
Author comments: Thank you for this summary.
2) Most sections of this paper are clearly written, but missing information about Mpox infection needs to be added to make the paper more informative for the topic it covers. Above all, there is a lack of detailed information on the multi-country Mpox outbreak in 2022 as a public health emergency on July 23, 2022, as well as on cases of Mpox infection in Pakistan.
Author comments: Thank you. We have added more information about multi-country Mpox outbreak in 2022 as well as cases reported in Pakistan in 2023. We trust this is now acceptable.
3) Comments (Major revision):
Line 70: It is mandatory to add a new paragraph that will describe in detail:
- Socio-demographic characteristics of patients and those who died in the multi-country outbreak of Mpox in 2022
- List the populations that were most at risk of Mpox in this outbreak
- List the risk factors for Mpox.
- List the classification according to the level of risk for Mpox infection for exposed persons.
- Cite appropriate references.
Author comments: Thank you for these comments. We have now added in additional details to those already recorded, and trust this is now acceptable.
4) Lines 105-111: The mentioned cases of Mpox infection in Pakistan were probably surveyed and provided their socio-epidemiological anamnesis. List all known risk factors for Mpox infection that have been identified in those cases. Pay attention to the way the infection is transmitted, personal medical history of the cases (immunosuppression, cancer, HIV, sexually transmitted diseases, etc.).
Author comments: Thank you. As documented, in the case of Pakistan, the majority of reported cases have a travel history from KSA and Somalia. We try to find information about local transmission of the cases but unfortunately no such information is readily available including from health authorities or hospitals. We hope this is acceptable to you.
5) The Methods section is satisfactorily written, including most of the subsections presented. As a whole, the appropriate methodology was applied in this paper.
Author comments: Thank you for these kind comments – appreciated
6) But - Lines 161-166: Explain why other questions were not included in the survey. Why did the authors not include in the survey other important questions about Mpox infection, based on their own clinical/epidemiological/public health knowledge and/or work with Mpox cases.
Author comments: Thank you. We have added where possible questions mainly in demographic sections that were relevant to the study population and study objectives. We hope this is OK.
7) The Results section presented the data of this research in a satisfactory manner.
Author comments: Thank you for these kind comments – appreciated
8) Lines 67-75. Other limitations of this study were not satisfactorily included. As limitations of this study, it is necessary to discuss the sample size, the incompleteness or absence of questions included in this survey about the characteristics of Mpox infection, there are no questions about risk factors and vaccines against Mpox infection.
Author comments: Thank you. We have updated limitation section, and hope this is now acceptable.
9) In summary: This work is not informative enough for the topic it deals with. Neither the word `risk factors' nor `risk exposure' for Mpox infection is mentioned in the paper. Information about this should be entered in the Introduction section. Without such information, about which no question was asked to the respondents in this study, as well as no question about the use of vaccines in the prevention of Mpox infection, this paper is not satisfactorily informative.
Author comments: Thank you. We have now added this into the introduction as well as highlighting details of the initial cases in Pakistan and activities within the Province to deal with cases including establishing dedicated wards in hospitals should be need arise. In addition, additional references regarding concerns with the knowledge of HCPs including physicians regarding Mpox. We trust this is now OK.

Reviewer 2 Report
Abstract:
The introduction of the abstract should be revised. I contains a number of no correlated sentences with outdated data (ex: "has recently"? It was in the last year., "Rising rates" ? It is currently in decline phase.
Line 35: correct : "of"
Line 36: add a point before 389
the majority of female: the majority were female
Line 37: you should provide the scale used. 17.69 from what? What can one understand from this number?
Introduction:
The introduction can be improved by summarizing and organizing the ideas
Results:
you should avoid to use adverbs in the results (encouragingly, ….). You should just describe your results. Adverbs should be used in the discussion
Correct the pages and lines numbering
Correct the items in the figures. Some of them are incomplete?
Improve the quality of your tables by putting the variables (age, gender… in a separate column).
Table 2: what did "a" and "b" mean?
Table 3: What did "B", "SE" and "Beta" mean? What did they represent exactly? What did -2.996 mean for example ? Also, develop more about the results of the regression analysis.
Discussion:
The first paragraph (lines 7-27). Summarize please.
You should include the results of the regression analysis and summarize the other parts that you included in your discussion.
At last, the manuscript should be checked for some language errors. (i.e: Our study revealed that majority of future HCWs are aware of the origin of Mpox in terms of its viral nature; however, a significant proportion of HCW participants were….(ou can write just:" ..proportion of them…".
Moderate editing of English language required
Author Response
Quality of English Language
( ) I am not qualified to assess the quality of English in this paper
( ) English very difficult to understand/incomprehensible
( ) Extensive editing of English language required
(x) Moderate editing of English language required
( ) Minor editing of English language required
( ) English language fine. No issues detected
Author comments: Thank you for this. The paper has now been updated by one of the co-authors who is a native English speaker with over 500 publications in peer-reviewed Journals since 2008. We trust this is now OK.
|
Yes |
Can be improved |
Must be improved |
Not applicable |
|
|
Does the introduction provide sufficient background and include all relevant references? |
( ) |
(x) |
( ) |
( ) |
|
Are all the cited references relevant to the research? |
(x) |
( ) |
( ) |
( ) |
|
Is the research design appropriate? |
( ) |
(x) |
( ) |
( ) |
|
Are the methods adequately described? |
( ) |
(x) |
( ) |
( ) |
|
Are the results clearly presented? |
( ) |
(x) |
( ) |
( ) |
|
Are the conclusions supported by the results? |
(x) |
( ) |
( ) |
( ) |
Comments and Suggestions for Authors
1) Abstract:
The introduction of the abstract should be revised. I contains a number of no correlated sentences with outdated data (ex: "has recently"? It was in the last year., "Rising rates" ? It is currently in decline phase.
Author comments: Thank you – now addressed. We trust this is now acceptable.
2) Line 35: correct : "of"
Author comments: Thank you – now addressed.
3) Line 36: add a point before 389
Author comments: Thank you – now addressed
4) the majority of female: the majority were female
Author comments: Thank you – now addressed.
5) Line 37: you should provide the scale used. 17.69 from what? What can one understand from this number?
Author comments: Thank you. We have added details into the updated paper and trust this is now OK.
6) Introduction: The introduction can be improved by summarizing and organizing the ideas
Author comments: Thank you for this comment. As seen – and with the help of other reviewers – we have started broadly about Mpox, its transmission, outbreaks and key factors including those associated with the 2022 outbreak. Subsequently commented on general prevention/ management before discussing Pakistan specific. Following this we discuss the key role of HCWs with information/ vaccination campaigns before looking at their general knowledge and why key also to investigate current knowledge among students HCWs – leading into this paper. We trust this is acceptable to you.
7) Results:
- i) you should avoid to use adverbs in the results (encouragingly, ….). You should just describe your results. Adverbs should be used in the discussion
Author comments: Thank you – now addressed
- ii) Correct the pages and lines numbering
Author comments: Thank you – this will be undertaken with the help of the Journal, with line numbering removed when and if the updated paper is finally published. We hope this is OK with you.
iii) Correct the items in the figures. Some of them are incomplete?
Author comments: Thank you. We have tried to improve this Figure given incomplete data. We will work with the Journal to update the Figures when and if the paper is accepted for publication.
- iv) Improve the quality of your tables by putting the variables (age, gender… in a separate column).
Author comments: Thank you. The quality of the tables are as per journal requirements. However, we will work with them to improve he Tables when and if the paper is accepted for publication.
- v) Table 2: what did "a" and "b" mean?
Author comments: Thank you. We have defined “a”, “b” and “c” in the first column of table, e.g. for age categories, ‘a’ represents ≤ 20 years, ‘b’ = 20-25 and ‘c’ represents students above the age of 25. Interpretation of all post-hoc tests are also given in the manuscript text. We trust this is now OK.
- vi) Table 3: What did "B", "SE" and "Beta" mean? What did they represent exactly? What did -2.996 mean for example ? Also, develop more about the results of the regression analysis.
Author comments: In regression analysis, “B” is the unstandardized beta coefficient and “SE” represents standard error of unstandardized beta. Beta represents the standardized beta value
The unstandardized beta represents the slope of the line between the predictor variable and the dependent variable. Consequently for Education variable which has three levels (medical, pharmacy and nursing), this would mean that for every one unit increase in this variable, the dependent variable (MPoX knowledge score) decreases by nearly 3 units.
Regarding the standard error for the unstandardized beta, the value is similar to the standard deviation for a mean. The larger the number, the more spread out the points are from the regression line. The more spread out the numbers are, the less likely that significance will be found.
Standardized beta works similarly to a correlation coefficient. It will range from 0 to 1 or 0 to -1, depending on the direction of the relationship. The closer the value is to 1 or -1, the stronger the relationship. With this, we can actually compare the variables to see which had the strongest relationship with the dependent variable, since all of them are on the 0 to 1 scale. In our case, Education was found to be the only factor that associated with dependent variable (strength of association was found to be moderately strong).
More information has been included about the regression analysis in the manuscript. We hope this is now OK.
8) Discussion:
- i) The first paragraph (lines 7-27). Summarize please.
Author comments: Thank you – now done
- ii) You should include the results of the regression analysis and summarize the other parts that you included in your discussion.
Author comments: Thank you. We have added results of regression analysis in the results section. Moreover, we have discussed the findings in the relevant section, and trust this is now acceptable.
9) At last, the manuscript should be checked for some language errors. (i.e: Our study revealed that majority of future HCWs are aware of the origin of Mpox in terms of its viral nature; however, a significant proportion of HCW participants were….(you can write just:" ..proportion of them…".
Comments on the Quality of English Language - Moderate editing of English language required
Author comments: Thank you for this. The paper has now been updated by one of the co-authors who is a native English speaker with over 500 publications in peer-reviewed Journals since 2008. We trust this is now OK.

Reviewer 3 Report
This study appears to be a valuable contribution to the understanding of healthcare students' knowledge levels regarding monkeypox in Pakistan. It addresses an important public health issue and highlights the role of education in improving preparedness for emerging infectious diseases among future healthcare workers. However, there are some issues and questions that should be addressed in the study.
1- The introduction section is fairly well organized but there are some parts that are redundant or need to be moved to results or discussion (e.g. line 99-109). Please try to trimmed the introduction section.
2- Why is it that parameters such as 'Family income (thousands)' and 'Types of institute' are missing from the baseline characteristic table in the research study, and can you explain why they were not included in the multivariate analysis?
3- They did not mention how the data collection tool and questionnaire were developed and if they test its validity.
4-Please have the entire manuscript proofread by a Fluent English speaker prior to resubmission to ensure all spelling and grammatical errors are corrected (for example knowledge oof monkeypox)
5- Some recent and relevant studies could enrich your manuscript
-https://doi.org/10.3390/vaccines11010019
-https://doi.org/10.1016/j.tmaid.2022.102533
-https://doi.org/10.1016/j.jiph.2022.11.004
Moderate editing of English language required
Author Response
Quality of English Language
( ) I am not qualified to assess the quality of English in this paper
( ) English very difficult to understand/incomprehensible
( ) Extensive editing of English language required
(x) Moderate editing of English language required
( ) Minor editing of English language required
( ) English language fine. No issues detected
Author comments: Thank you for this. The paper has now been updated by one of the co-authors who is a native English speaker with over 500 publications in peer-reviewed Journals since 2008. We trust this is now OK.
|
Yes |
Can be improved |
Must be improved |
Not applicable |
|
|
Does the introduction provide sufficient background and include all relevant references? |
( ) |
(x) |
( ) |
( ) |
|
Are all the cited references relevant to the research? |
( ) |
(x) |
( ) |
( ) |
|
Is the research design appropriate? |
( ) |
(x) |
( ) |
( ) |
|
Are the methods adequately described? |
( ) |
(x) |
( ) |
( ) |
|
Are the results clearly presented? |
( ) |
(x) |
( ) |
( ) |
|
Are the conclusions supported by the results? |
( ) |
(x) |
( ) |
( ) |
Comments and Suggestions for Authors
1) This study appears to be a valuable contribution to the understanding of healthcare students' knowledge levels regarding monkeypox in Pakistan. It addresses an important public health issue and highlights the role of education in improving preparedness for emerging infectious diseases among future healthcare workers. However, there are some issues and questions that should be addressed in the study.
Author comments: Thank you for your positive comments and suggestions – appreciated! We hope we have adequately addressed the comments made.
2) The introduction section is fairly well organized but there are some parts that are redundant or need to be moved to results or discussion (e.g. line 99-109). Please try to trimmed the introduction section.
Author comments: Thank you for this. We have trimmed the Introduction where we can. However, we do believe that certain sections are key to lay the foundation for the study and trust this is acceptable to you. In addition – further points have been added following the requests from other Reviewers, as well as the additional references you kindly provided. We also hope this is acceptable to you.
3) Why is it that parameters such as 'Family income (thousands)' and 'Types of institute' are missing from the baseline characteristic table in the research study, and can you explain why they were not included in the multivariate analysis?
Author comments: All study participants belonged to public institutes, therefore, comparisons of Mpox knowledge score couldn’t be made between public and private sector students. We have added this in the limitation. Regarding “family income”, we have included analysis of this variable in the Tables as well in the manuscript text, and hope this is now acceptable.
4) They did not mention how the data collection tool and questionnaire were developed and if they test its validity.
Author comments: Thank you for this comment. Details of questionnaire development and its validation are in fact described in section 2.2 (survey instrument). We trust this is now OK.
5) Please have the entire manuscript proofread by a Fluent English speaker prior to resubmission to ensure all spelling and grammatical errors are corrected (for example knowledge oof monkeypox)
Author comments: Thank you for this. The paper has now been updated by one of the co-authors who is a native English speaker with over 500 publications in peer-reviewed Journals since 2008. We trust this is now OK.
6) Some recent and relevant studies could enrich your manuscript
-https://doi.org/10.3390/vaccines11010019
-https://doi.org/10.1016/j.tmaid.2022.102533
-https://doi.org/10.1016/j.jiph.2022.11.004
Author comments: Thank you – now incorporated.

Reviewer 4 Report
Please see file attached

Author Response
Quality of English Language
( ) I am not qualified to assess the quality of English in this paper
( ) English very difficult to understand/incomprehensible
( ) Extensive editing of English language required
( ) Moderate editing of English language required
( ) Minor editing of English language required
(x) English language fine. No issues detected
Author comments: Thank you – appreciated
|
Yes |
Can be improved |
Must be improved |
Not applicable |
|
|
Does the introduction provide sufficient background and include all relevant references? |
(x) |
( ) |
( ) |
( ) |
|
Are all the cited references relevant to the research? |
(x) |
( ) |
( ) |
( ) |
|
Is the research design appropriate? |
(x) |
( ) |
( ) |
( ) |
|
Are the methods adequately described? |
(x) |
( ) |
( ) |
( ) |
|
again |
(x) |
( ) |
( ) |
( ) |
|
Are the conclusions supported by the results? |
( ) |
(x) |
( ) |
( ) |
1) Overall Comments: Thank you for the opportunity to review this article titled “Knowledge of Human Monkeypox Infection among Final Year Medical, Pharmacy and Nursing Students: A Multicentre, Cross-sectional Analysis from Pakistan”. This research is highly relevant and important as it provides valuable insights into the knowledge of monkeypox among future healthcare workers (HCWs). It provides the first data on the knowledge of monkeypox among future HCWs in Pakistan. The authors identifie the factors that are associated with knowledge of monkeypox, such as age, gender, and education. In addition, it highlights the need for educational activities to improve monkeypox-related knowledge among future HCWs. It suggests that emerging infectious diseases, such as monkeypox, should be routinely incorporated into HCW curricula. This work is an important contribution to the field of public health and will help to improve the preparedness of HCWs to respond to future monkeypox outbreaks.
Author comments: Thank you for these kind comments – appreciated!
2) Specific Section Comments Introduction and Hypothesis: The introduction is informative and well-written. It provides the reader with a good understanding of the monkeypox virus and its importance to HCWs. The authors have also clearly stated the purpose of their study and how the findings will be used. They also discuss the role of HCWs in preventing and managing monkeypox, as well as the need to educate healthcare students about the virus.
It was interesting to read about the emergence of monkeypox through the years. In addition, the authors discuss the fact that monkeypox is a zoonotic disease, and explained the meaning where it can be transmitted from animals to humans. They made the readers aware of the importance for HCWs to know about this, as they may come into contact with infected animals during their work.
The authors highlight the lack of studies on the knowledge of monkeypox among HCWs in Pakistan. This is a significant gap in knowledge, and the authors' proposed that their study will help to fill this gap.
Author comments: Thank you again for these kind comments – appreciated!
3) Materials and Methods: The methodology section provides a comprehensive description of the study design, setting and data collection process. The use of a cross-sectional, questionnaire-based study design is appropriate, and there is good justification for the research setting (the Punjab province of Pakistan and public sector universities including 3 medical, 2 pharmacy and 6 nursing).
It was good to see that content validation of the survey was performed by a panel of seven experts from the field of nursing, pharmacy and medicine. Addition rigor of study design was demonstrated by the pilot survey the was conducted with 30 students. The authors included accurate details about the demographics and characteristics of the study participants, such as name of the institution, age, sex, residence and family income, which was valuable for readers to understand the study's population.
The questionnaire/ survey was well supported by the Bloom’s cut-off criteria, that has been used in several previously cited articles. The sample size was appropriate and the authors even recruited up 5% more participants to compensate for incomplete data sets.
Author comments: Thank you again for these kind comments – appreciated!
4) Results and Data Analysis: The presentation of the study results is clear with appropriate statistical analysis. Linear regression analysis was suitable to confirm the prediction of Mpox knowledge with education. It was interesting although not surprising to see that students with medical and pharmacy education possessed better knowledge than nursing students, considering the intensity and duration of their first degrees.
The authors clearly showed in table 3, that there were other factors (apart from educating) that significantly affected participants knowledge on Mpox. The reviewer commends the authors for the simplicity of this table. Discussion and Conclusion: Generally, the discussion section of the article is a valuable contribution to the literature on monkeypox.
The authors effectively interpret the study's results and relates them to existing literature. The authors state that this is the first study to evaluate the knowledge of future HCWs in Pakistan about monkeypox. This is a significant contribution, as it provides much-needed data on the knowledge gaps in this important area. There was appropriate discussion on the findings of other studies about the knowledge of monkeypox among HCWs. The authors noted that their findings are generally consistent with those of other studies, although they do find that their participants are more knowledgeable about the transmission of the virus than participants in some other studies. This may be interesting to journal readers. The conclusions drawn are well-supported by the evidence presented in the results. And overall, the discussion section of the article is well-written and informative.
Author comments: Thank you again for these kind comments – appreciated!
5) There are however a few areas for improvement:
- i) First, a clear heading of limitations would be useful. Here, the authors could provide more detail about the limitations of their study. For example, they could discuss the fact that they only collected data from public sector educational institutions in Punjab, and that they used convenience sampling. This was previously mentioned in the main text, but it would be beneficial to add it in this appropriate section. This would help readers to better understand the limitations of the study and the generalizability of the findings.
Author comments: Thank you – now inserted.
- ii) Second, the authors could discuss the implications of their findings in more detail. For example, they could discuss how the findings could be used to improve the prevention and management of monkeypox outbreaks. This would help readers to see the practical applications of the study's findings.
Author comments: Thank you. We have added where needed the implications of our study. We hope it will be accepted now.

Round 2
Reviewer 1 Report
Thanks for the opportunity to re-review the manuscript ID: healthcare-2621169.
The authors responded to my comments and made certain corrections in the revised version of this manuscript. Also, the authors have provided appropriate explanations for some issues.
Thanks to the authors.
The quality of English language is appropriate.
Reviewer 2 Report
The authors have answered to all my comments
Minor editing of English language required
Reviewer 3 Report
I am satisfied that the authors have addressed all of my previous concerns about the article. It is now much improved and I feel that it is now suitable for publication.
Minor editing of English language required